# Exploring the Role of Metabolic Hormones in Amyotrophic Lateral Sclerosis

**DOI:** 10.3390/ijms25105059

**Published:** 2024-05-07

**Authors:** Anca Moțățăianu, Ion Bogdan Mănescu, Georgiana Șerban, Laura Bărcuțean, Valentin Ion, Rodica Bălașa, Sebastian Andone

**Affiliations:** 1Department of Neurology, University of Medicine, Pharmacy, Science and Technology of Târgu Mureș ‘George Emil Palade’, 540142 Târgu Mureș, Romania; 21st Neurology Clinic, Mures County Clinical Emergency Hospital, 540136 Târgu Mureș, Romania; 3Department of Laboratory Medicine, University of Medicine, Pharmacy, Science and Technology of Târgu Mureș ‘George Emil Palade’, 540142 Târgu Mureș, Romania; 4Doctoral School, University of Medicine, Pharmacy, Science and Technology of Târgu Mureș ‘George Emil Palade’, 540142 Târgu Mureș, Romania; 5Faculty of Pharmacy, Department of Analytical Chemistry and Drug Analysis, University of Medicine, Pharmacy, Science and Technology of Târgu Mureș ‘George Emil Palade’, 540142 Târgu Mureș, Romania; 6Drug Testing Laboratory, University of Medicine, Pharmacy, Science and Technology of Târgu Mureș ‘George Emil Palade’, 540142 Târgu Mureș, Romania

**Keywords:** ALS, amyotrophic lateral sclerosis, metabolic hormones, insulin, amylin, GLP-1, ALSFRS

## Abstract

Amyotrophic lateral sclerosis (ALS) is a devastating neurodegenerative disease characterized by progressive loss of motor neurons. Emerging evidence suggests a potential link between metabolic dysregulation and ALS pathogenesis. This study aimed to investigate the relationship between metabolic hormones and disease progression in ALS patients. A cross-sectional study was conducted involving 44 ALS patients recruited from a tertiary care center. Serum levels of insulin, total amylin, C-peptide, active ghrelin, GIP (gastric inhibitory peptide), GLP-1 active (glucagon-like peptide-1), glucagon, PYY (peptide YY), PP (pancreatic polypeptide), leptin, interleukin-6, MCP-1 (monocyte chemoattractant protein-1), and TNFα (tumor necrosis factor alpha) were measured, and correlations with ALSFRS-R, evolution scores, and biomarkers were analyzed using Spearman correlation coefficients. Subgroup analyses based on ALS subtypes, progression pattern of disease, and disease progression rate patterns were performed. Significant correlations were observed between metabolic hormones and ALS evolution scores. Insulin and amylin exhibited strong correlations with disease progression and clinical functional outcomes, with insulin showing particularly robust associations. Other hormones such as C-peptide, leptin, and GLP-1 also showed correlations with ALS progression and functional status. Subgroup analyses revealed differences in hormone levels based on sex and disease evolution patterns, with male patients showing higher amylin and glucagon levels. ALS patients with slower disease progression exhibited elevated levels of amylin and insulin. Our findings suggest a potential role for metabolic hormones in modulating ALS progression and functional outcomes. Further research is needed to elucidate the underlying mechanisms and explore the therapeutic implications of targeting metabolic pathways in ALS management.

## 1. Introduction

Amyotrophic lateral sclerosis (ALS) is a heterogeneous, progressive neurodegenerative disease with a multifactorial potential specific pathogenic mechanism leading to irreversible motor neuron (MN) degeneration from the brain cortex and spinal cord. The pathogenesis of ALS is a complex process, with interlinked and overlapping pathways such as oxidative stress, mitochondrial dysfunction, neuroinflammation, and glutamate excitotoxicity, all of which appear to be contributory [1,2]. Despite extensive research efforts, effective treatment for ALS remains elusive, and the disease continues to have a devastating impact on patients and their families [3,4].

In recent years, there has been growing interest in the potential role of metabolic hormones in ALS pathogenesis and progression. Metabolic hormones, such as insulin, amylin, leptin, and others, play critical roles in regulating various physiological processes, including glucose metabolism, energy homeostasis, and inflammation. Emerging evidence suggests that dysregulation of these hormones could contribute to the neurodegenerative process in ALS, mainly due to the high energy demands of MNs [5]. Studies indicate that dysfunctional energy metabolism may significantly influence disease progression, with increased resting energy expenditure and decreased BMI associated with poorer prognosis [6]. Furthermore, disturbances in energy metabolism have been observed in ALS patients before motor symptom onset, including alterations in BMI, energy intake, and fat consumption [7,8]. 

Brain insulin signaling has accumulated significant attention in neuroscience research. While insulin is primarily recognized for regulating blood glucose levels, it serves numerous additional functions in physiology. Growth factors such as insulin, GLP-1, and GIP can cross the blood–brain barrier (BBB), activating receptors expressed by neuronal and glial cells. The activation of insulin receptors from the central nervous system (CNS) triggers critical signaling pathways that stimulate various processes, including cell metabolism, synaptic activity, glucose utilization, and gene expression related to oxidative stress response, as well as genes involved in cell repair and growth [9,10,11]. Insulin, GLP-1, and GIP also reduce brain inflammation by acting on microglia and astroglia receptors, reducing pro-inflammatory cytokine levels and modulating neuroinflammation [12,13]. Furthermore, the activation of GIP/GLP-1 receptors enhances microglial expression of growth factors like brain-derived neurotrophic factor (BDNF), nerve growth factor (NGF), and glial cell line-derived neurotrophic factor (GDNF) [14,15].

Our study aimed to investigate the relationship between metabolic hormones and disease progression in ALS patients. We comprehensively analyzed metabolic hormone levels in ALS patients compared to healthy controls and explored their associations with disease severity, progression patterns, and clinical outcomes. Our findings highlight the intricate interplay between metabolic hormones and neurodegeneration in ALS and underscore the potential utility of metabolic hormones as biomarkers or therapeutic targets for ALS. 

With a complex etiology and limited treatment options, metabolic dysregulation has emerged as a potential contributor to ALS pathogenesis, yet its precise role remains elusive. In this article, we present the results of our study, providing valuable insights into the complex relationship between metabolic hormones and ALS progression. By elucidating these relationships, we aim to contribute to the growing knowledge surrounding ALS pathogenesis and identify new pathways for therapeutic intervention.

## 2. Results

### 2.1. General Groups’ Characteristics

Our studied groups contained 44 ALS patients with an age of 58.05 ± 12.14 and 40 control patients with an age of 56.97 ± 11.83. There was no statistically significant difference between the two groups regarding age, proving that our controls were age-matched (*p* = 0.6813). The sex ratio was almost the same between the two, and the controls were also sex-matched.

The other clinical characteristics of ALS patients can be found in Table 1.

### 2.2. Metabolic Hormones: Control vs. ALS

Compared with the control group, the ALS patients had significantly higher serum values for amylin, GIP, glucagon, insulin, and PP and significantly lower serum values for C-peptide, ghrelin, and leptin.

The only hormones without significant differences between the two groups were GLP-1, MCP-1, and PYY.

All serum values for control and ALS patients can be found in Table 2.

### 2.3. Clinical, Paraclinical, and Metabolic Hormones’ Correlations

All the bivariate correlations between metabolic hormones and ALS evolution scores and biomarkers are summarized in Table 3.

However, some correlations required special attention.

From all metabolic hormones, insulin and total amylin values significantly correlated with the ALS evolution scales.

Insulin had a highly significant negative correlation with DPR (*p* = 0.001, r = −0.303) while having an extremely significant positive correlation with ALSFRS-R total score (*p* < 0.001, r = 0.394), as well as with ALSFRS-R subscores like Respiratory (*p* < 0.001, r = 0.411), Bulbar (*p* = 0.001, r = 0.307), and Fine Motor (*p* < 0.001, r = 0.341). No correlation was found regarding the Gross Motor subscore.

Insulin also had several positive correlations with other metabolic hormones, such as amylin (*p* < 0.001, r = 0.748), C-peptide (*p* < 0.001, r = 0.606), GIP (*p* < 0.001, r = 0.482), leptin (*p* = 0.002, r = 0.299), and PP (*p* < 0.001, r = 0.305).

Amylin followed a similar trend to insulin, being negatively correlated with DPR (*p* = 0.016, r = −0.230) and positively correlated with the Respiratory subscore (*p* = 0.006, r = 0.260), Bulbar subscore (*p* = 0.013, r = 0.237), and Fine Motor subscore (*p* = 0.002, r = 0.288), as well having a highly significant positive correlation with the Total score (*p* < 0.001, r = 0.341). The exact missing correlation with the Gross Motor subscore was found for amylin.

Although amylin had the same correlations as insulin, the strength and significance of the correlations were much stronger for insulin than for amylin.

Amylin had a positive correlation with other metabolic hormones, like C-peptide (*p* < 0.001, r = 0.529), GIP (*p* < 0.001, r = 0.493), leptin (*p* < 0.013, r = 0.237), and PP (*p* = 0.001, r = 0.300).

Other metabolic hormones that negatively correlated with DPR were C-peptide (*p* = 0.008, r = −0.255) and leptin (*p* = 0.023, r = −0.217). No other hormones seemed to have any significant correlation with DPR.

As for the ALSFRS-R total score, only GLP-1 had a significant positive correlation (*p* = 0.032, r = 0.205).

However, for ALSFRS-R subscores, there were other specific correlations, such as leptin’s significant positive correlation with the Fine Motor subscore (*p* = 0.028, r = 0.210).

Another correlation worth mentioning is the MCP-1, the only metabolic hormone that had a negative correlation with ALSFRS Respiratory subscore (*p* = 0.019, r = −0.224) and Bulbar subscore (*p* = 0.004, r = −0.274).

Other correlations between the metabolic hormones were found, as reported in Table 3. Most of them were positive correlations between different metabolic hormones, except MCP-1, which had a negative correlation with GLP-1 alone (*p* = 0.003, r = −0.280).

Diffusion time and generalization times were positively correlated with GLP-1 and Glucagon but again negatively correlated with MCP-1.

BMI had positive correlations with amylin (*p* = 0.001, r = 0.306), C-peptide (*p* = 0.020, r = 0.222), GLP-1 (*p* = 0.018, r = 0.225), insulin (*p* < 0.001, r = 0.340), and leptin (*p* < 0.001, r = 0.590).

The FVC(%) had extremely significant positive correlations with insulin (*p* < 0.001, r = 0.340) but also a negative correlation with MCP-1 (*p* = 0.010, r = −0.245).

### 2.4. Subgroup Analysis: Based on Sex, ALS Subtype, Progression, and Evolution Patterns

The sex subgroup analysis showed significant statistical differences between metabolic hormones’ serum values.

Both amylin and glucagon had higher values in male patients (354.28 ± 418.06, respectively, 80.58 ± 55.65) than in female patients (160.026 ± 156.5684, respectively, 58.61 ± 32.23), and these differences were statistically significant (*p* = 0.006, respectively, *p* = 0.026).

However, leptin showed the opposite; the values in male patients (3303.72 ± 2804.06) were lower than in female patients (7136.61 ± 5225.67) with extreme significance (*p* < 0.001).

The ALS subtypes’ subgroup analysis did not reveal any statistically significant differences in values of metabolic hormones.

Only glucagon showed significant differences (*p* = 0.027) in the pattern of evolution, as the horizontal pattern had lower serum values (63.27 ± 36.71) than the vertical pattern (84.22 ± 60.09).

When we split the patients based on the speed of evolution, using a cut-off value of 0.67 for DPR as a biomarker index, we observed that patients with slower evolutionary forms had almost twofold increased values for amylin (*p* = 0.014) and insulin (*p* = 0.007). The specific values can be observed in Table 4 as well as in Figure 1.

## 3. Discussion

Our study reveals several important correlations between metabolic hormones and various aspects of ALS clinical outcome and progression. Our data indicate a potential role for insulin and amylin in modulating ALS progression. Notably, insulin exhibits a negative correlation with the disease progression rate; higher insulin levels were correlated with slower disease progression and higher functional outcomes, assessed by ALSFRS-R total score and subscores for respiratory function, bulbar function, and fine motor skills, suggesting a protective effect on MNs’ function. Similar correlations were observed for amylin but with weaker strength and significance than insulin. These findings suggest a potential role for insulin and amylin in modulating ALS progression and functional outcomes.

### 3.1. Role of Insulin and Glucose Metabolism in ALS

Increasing evidence indicates that decreased brain metabolism, impaired metabolic hormone signaling, and resistance are associated with the progression of neurodegenerative disorders, including ALS [16]. Disruptions in glucose metabolism and insulin signaling pathways are significant factors in ALS pathogenesis. They impact various aspects crucial for neuronal health, including energy metabolism, mitochondrial function, inflammatory responses, and ultimately neuronal survival. ALS patients often experience impaired energy metabolism, marked by hypermetabolism, dysfunctional mitochondria, and insufficient energy production. These disruptions contribute to the disease’s progression, leading to poor prognosis and weight loss in affected individuals [17,18,19]. 

The neuroprotective effects of insulin in ALS are a subject of research interest due to the potential therapeutic implications in this ‘no hope’ and devastating disease. While the exact mechanisms are still being elucidated, several hypotheses suggest how insulin may exert beneficial effects in ALS, including the modulation of glutamate excitotoxicity, support of mitochondrial function, anti-apoptotic effects, promotion of neurotrophic signal, and modulation of glucose metabolism [9,11].

Epidemiological studies have demonstrated that type 2 diabetes (T2D), characterized by hyperinsulinemia, is protective against ALS. In contrast, type 1 diabetes (T1D), characterized by a total lack of insulin, is associated with an increased risk of ALS. Two population-based studies conducted on the Scandinavian population from Denmark and Sweden found that while T2D was linked to a decreased risk of ALS, no such association was observed for T1D. In the Swedish population, an inverse relationship between ALS and T2D was particularly notable six years following T2D onset [20,21]. In an Italian cohort study, a markedly reduced risk of ALS was observed in individuals with T2D, irrespective of sex, age, or ALS phenotype [22]. Additionally, Zhang et al. reported a significant decrease in the risk of ALS associated with genetically predicted T2D in both European and East Asian populations [23]. 

Although very few studies have compared the level of metabolic hormones in ALS patients and controls, Bilic et al. studied the differences in insulin, growth hormone, and insulin growth factor-1 (IGF-1) levels in ALS patients and healthy controls. They sampled serum and cerebrospinal fluid probes and found that insulin decreased in ALS patients and controls. Even if these results contradict our finding, which showed significantly increased serum values in ALS patients compared to control, it is essential to note that abnormal insulin values seem to be associated with ALS [24].

However, Pradat et al. showed that patients with ALS presented impaired glucose tolerance and had higher insulin serum values compared to the control group, similar to our study groups. They noted, though, that insulin resistance was not correlated with disease duration or severity without being able to elucidate whether insulin resistance is a cause or a consequence of the neurodegeneration that occurs in ALS [25]. 

The highly insulin-sensitive brain relies on insulin to regulate metabolism and normal CNS function [26]. Insulin crosses the blood–brain barrier (BBB) via a saturable transport process, primarily binding to insulin receptors (IR) and insulin-like growth factor-1 receptor (IGF-1R), abundant in key CNS regions like the olfactory bulb, hypothalamus, and cerebral cortex [27,28,29]. After insulin binds to IR/IGF-1, it triggers signaling cascades involving phosphatidylinositol 3-kinase (PI3K), protein kinase B (Akt), and mammalian target of rapamycin (mTOR), influencing neuronal and glial growth, repair, survival, and synaptic plasticity [29,30,31]. Insulin’s multifaceted role extends to regulating overall brain function and whole-body energy balance [11,32,33,34,35].

ALS is a complex disease caused by various interacting pathological changes. Despite extensive research, particularly in sporadic cases, the precise causative mechanism remains to be determined. Evidence suggests that ALS results from neuroinflammation, oxidative stress, mitochondrial dysfunction, glutamate excitotoxicity, and impaired energy metabolism [1,36]. Impaired energy metabolism plays a significant role in ALS progression, possibly preceding MN loss and disease onset [7,19,37,38,39].

Glutamate excitotoxicity, a prominent feature of ALS, involves the activation of ionotropic glutamate receptors (IGluRs), such as AMPA (α-amino-3-hydroxy-5-methyl-4-isoxazolepropionic acid) and NMDA (N-methyl-D-aspartate) receptors. This activation leads to a toxic influx of extracellular Na^+^ and Ca^2+^, contributing to the degeneration of motor neurons (MNs). A recognized negative correlation between neurodegeneration and synaptic plasticity is influenced mainly by iGluRs’ function [40]. AMPA receptors’ high expression on postsynaptic terminals reduces their ability to buffer Ca^2+^, thereby contributing to MN degeneration due to Ca^2+^-induced excitotoxicity [41]. Insulin induces the endocytosis of AMPA receptors, reducing their surface expression, and modulates NMDA receptors, enhancing their activity for synaptic strengthening. This modulation of glutamatergic receptors allows insulin to influence synaptic activity and plasticity, providing neuroprotective effects [42,43,44].

Connexin 43 (Cx43) channel proteins, mainly expressed in astrocytes, form pore-like hemichannels facilitating the diffusion of molecules and ions, including excitatory amino acids, to MNs, leading to Ca^2+^ overload [45,46]. During disease progression, Cx43 is expressed in the spinal cord and motor cortex of SOD1G93A ALS mouse models. This upregulation is associated with heightened hemichannel activity and gap junction coupling, resulting in elevated intracellular Ca^2+^ concentration, contributing to glutamate excitotoxicity and ALS progression [47,48]. Additionally, Cx43 mediates the interaction between activated microglia and astrocytes during advanced ALS stages, potentially influencing microglial reactivity and neuroinflammation [49]. Post-mortem studies on ALS patients reveal elevated Cx43 expression, particularly in those with rapid disease progression. Depletion of Cx43 or Cx43 knockout from astrocytes extends survival in SOD1-ALS mouse models. Tonabersat, a Cx43 blocker drug, preserves motor neurons in SOD1-G93A mutant mouse models [50]. Recently, Lehrer et al. demonstrated through in silico modeling that insulin can bind to and block Cx43, inhibiting the release and transfer of toxic molecules like glutamate, potentially slowing ALS progression [51].

Mitochondrial dysfunction is a hallmark of ALS, characterized by ATP deficiency, ROS overproduction, inflammation, and increased proapoptotic molecules, contributing to programmed cell death [52,53]. Oxidative phosphorylation, the primary ATP production process, occurs exclusively in mitochondria [54]. Brain insulin/IGF-1 signaling influences mitochondrial function, affecting biogenesis and oxidative capacity. Insulin deficiency or resistance decreases muscle mitochondrial ATP production and oxidative phosphorylation gene expression, while insulin infusion enhances mitochondrial protein expression, oxidative enzyme activity, and ATP synthesis in muscles [55,56]. IGF-1 protects mitochondria from apoptosis and upregulates mitophagy in ALS murine and cell models [57]. 

Insulin treatment prevented oxidative stress-induced decreases in intracellular glutathione levels in cortical neurons by activating insulin-mediated signaling pathways, including PI-3K. Insulin also stimulated glutathione reductase activity and inhibited glutathione peroxidase activity, indicating modulation of the glutathione redox cycle. These findings suggest insulin’s potential in preventing oxidative stress-related injury [58].

IGF-1, a hormone structurally similar to insulin, exhibits potent neurotrophic effects. Insulin can elevate IGF-1 levels and activate downstream signaling pathways, promoting neuronal survival and plasticity [29]. Nagel et al. investigated the association between serum IGF-1 levels and ALS risk and prognosis. They found that higher IGF-1 levels were associated with improved prognosis and survival in ALS patients, indicating a potential role for IGF-1-related mechanisms in ALS progression and survival [59].

Insulin plays a crucial role in brain glucose metabolism through several mechanisms: (1) promoting glucose transport, insulin facilitates glucose uptake into brain cells by promoting the translocation of glucose transporter proteins such as GLUT3 in neurons and GLUT1 in astrocytes to the cell membrane; (2) enhancing glycolysis, insulin activates key enzymes involved in glycolysis, thereby increasing glycolytic flux to produce ATP, the primary energy source for neurons and glial cells; and (3) producing energy, glycolysis converts glucose to pyruvate in the cytosol, generating ATP and nicotinamide adenine dinucleotide (NADH), which are essential for maintaining neuronal excitability, synaptic transmission, and other energy-demanding processes in the CNS.

Improving glucose metabolism produces ATP and may reduce abnormal protein aggregation and oxidative stress, critical factors in ALS progression. ATP’s hydrotrope property inhibits protein aggregate formation and breaks down existing aggregates. Targeting the glycolysis pathway via metabolic therapy can attenuate ALS progression and improve survival [60,61].

In a population-based, case–control study, Mariosa et al. used the data from a 20-year period regarding the Swedish population consisting of over 5000 patients with ALS. The objective of the study was to find any association between diabetes and ALS. They found that there is an inverse association between diabetes and ALS risk (OR 0.66), especially in non-insulin-dependent patients. This association was age-dependent and was found only in patients over 70 years old. What is more interesting is that the data for younger patients than 50 years old showed an increased ALS risk (OR 5.38), especially for insulin-dependent patients. Given the fact that most patients over 70 years old who are non-insulin-dependent have type 2 diabetes, while young patients with insulin-dependent diabetes have type 1 diabetes, this means that diabetes can also be a protective factor but also a risk factor for ALS depending on the type. While type 1 diabetes seems to increase the chance for ALS incidence, type 2 diabetes reduces it. Therefore, their study demonstrated that there is a clear association between diabetes and ALS, but we have to take into consideration other variables as well such as age and insulin dependence [20].

### 3.2. Role of Amylin in ALS

Our study results suggest that amylin may influence disease progression and functional status in ALS. Elevated amylin levels correlate with slower disease progression and improved respiratory function, bulbar function, and fine motor skills in ALS patients. Amylin, a pancreatic satiation peptide hormone co-secreted with insulin, typically exists as a soluble monomer. However, in T2D, it can become insoluble and aggregate, aberrantly potentially linking T2D with neurodegenerative diseases such as Alzheimer’s disease. [62,63]. Preclinical Alzheimer’s studies demonstrate amylin’s neuroprotective effects in modulating oxidative stress and neuroinflammation, particularly through synthetic analogues like pramlintide [64,65,66]. Limited data exist on amylin’s role in ALS, so our study contributes significant insights into the role of amylin in ALS, filling a crucial gap in the current understanding of this topic. 

Subgroup analysis based on sex revealed significant differences in metabolic hormone levels, with higher amylin and glucagon levels in male patients and higher leptin levels in female patients. However, ALS subtype analysis (spinal vs. bulbar) did not reveal significant differences in metabolic hormone levels. Regarding the pattern of disease evolution, glucagon levels showed significant differences between horizontal and vertical patterns, suggesting a potential association between hormone levels and the pattern of disease progression. Finally, patients with slower disease evolution exhibited higher levels of amylin and insulin compared to those with faster disease evolution, highlighting potential associations between hormone levels and disease progression rates.

### 3.3. Role of GLP-1 in ALS

GLP-1 (glucagon-like peptide-1), an incretin hormone, regulates glucose homeostasis and insulin secretion, sharing growth factor-like properties with insulin. Crossing the BBB, GLP-1 activates its receptor (GLP-1R) in the brain, acting as a neurotrophic factor to protect neurons from toxic effects. GLP-1R signaling pathways exhibit neuroprotective effects, supporting neurogenesis and synaptic plasticity, reducing inflammation and oxidative stress, and inhibiting apoptosis [13,67,68,69,70]. 

In our study, we observed negative correlations between metabolic hormones such as C-peptide and leptin and the rate of ALS progression, suggesting potential protective effects against disease progression. Additionally, GLP-1 showed a significant positive correlation with the ALSFRS-R total score, indicating a beneficial impact on the patient’s functional status and excellent motor skills. Furthermore, we investigated associations between metabolic hormones and diffusion and generalization times, revealing positive correlations with GLP-1 and glucagon and negative correlations with MCP-1. These findings suggest a potential influence of GLP-1 and glucagon on disease spread and generalization in ALS, linked to slower neurodegeneration and delayed progression of ALS. Interestingly, MCP-1 was the only hormone negatively correlated with ALSFRS-R Respiratory and Bulbar subscores, suggesting potentially detrimental effects on ALS patients’ functional domains.

Potential roles of GLP-1 signaling activators in neurodegenerative diseases have shown the neuroprotective effects of GLP-1 and GIP agonists in Alzheimer’s and Parkinson’s patients and murine models. Dual GLP-1/GIP receptor agonists offer superior neuroprotection compared to single GLP-1 analogues. Recent advancements include engineered dual analogues to penetrate the BBB, reducing inflammation, oxidative stress, and apoptosis [71]. Li et al. demonstrated neurotrophic and neuroprotective effects against oxidative stress and apoptosis through GLP-1R activation in cellular and murine models of ALS [72]. The GLP-1 pathway may play a crucial role in ALS by facilitating neurogenesis and neuroplasticity, with GLP-1 activators showing promise as targets for neuroprotection in ALS to slow disease progression and improve patient survival in neurodegenerative diseases, but further studies are needed [73].

### 3.4. Role of Other Metabolic Hormones in ALS

Hubbard et al. showed increased glucagon levels in a small ALS patient group, potentially explaining observed glucose intolerance, abnormal insulin reactions, and plasma amino acid level abnormalities [74]. Understanding glucagon and GLP-1 influence in ALS is limited, and further studies are needed to investigate if dysregulated glucagon secretion contributes to abnormal glucose homeostasis and influences the disease progression.

In our research, MCP-1 (monocyte chemoattractant protein-1) exhibited negative associations with ALSFRS-R, Respiratory, and Bulbar subscores, suggesting a link between higher MCP-1 levels and poorer respiratory and bulbar function in ALS. This negative correlation implies that elevated MCP-1 levels may worsen respiratory and bulbar symptoms, significant contributors to overall functional impairment in ALS. Additionally, MCP-1 negatively correlated with GLP-1, an incretin hormone regulating glucose homeostasis and insulin secretion, suggesting a potential interaction between inflammatory processes (indicated by MCP-1 levels) and metabolic regulation (represented by GLP-1 levels). This finding underscores the intricate interplay between neuroinflammation and metabolic dysregulation in ALS. Martinez et al. demonstrated increased MCP-1 and MIP-1β levels in the cerebrospinal fluid of 77 ALS patients, suggesting a synergistic effect of these cytokines in ALS pathogenesis. However, no differences in MCP-1 levels were found concerning ALS duration or disease severity [75].

Ghrelin and leptin were among the hormones found to be significantly reduced in the ALS group compared to controls in our study. Both have been considered potential therapies for ALS. Ngo et al. suggested ghrelin’s potential based on its role in regulating food intake, body weight, and cellular metabolism, noting an association between body weight loss and faster disease progression in ALS patients [76]. Our results supported this assumption, showing decreased serum ghrelin levels in ALS patients. Similarly, leptin has shown promise in experimental studies on murine models, where administering leptin improved disease duration and motor performance [77]. In our research, we also observed decreased leptin levels in ALS patients compared to controls, with leptin negatively correlating with the rate of ALS progression, suggesting potential neuroprotective effects. Nagel et al. found similar results, with lower leptin levels in ALS patients associated with a higher risk of ALS and poorer survival outcomes, particularly in men [78]. Evidence suggests that leptin may exert neuroprotective effects through anti-inflammatory actions, the modulation of oxidative stress and mitochondrial function, promoting neurogenesis and synaptic plasticity, and regulating energy homeostasis [79]. Understanding leptin’s mechanisms in neuronal health may offer new insights into neurodegenerative disease pathogenesis and identify therapeutic targets for intervention.

Furthermore, BMI positively correlated with several metabolic hormones, indicating potential interactions between metabolic status and hormone levels in ALS patients. The relationship between BMI and metabolic hormones in ALS patients is complex and multifactorial, involving interactions among adipose tissue, insulin resistance, inflammation, and hormonal signaling pathways [80,81]. Several articles have examined the relationship between BMI and ALS. In one such study, the authors found an inverse correlation between the decline of ALSFRS-R and the BMI of patients, indicating that individuals with a higher BMI experienced a slower progression rate of ALSFRS-R [82].

In our study, these data will be the subject of a future article focusing on the follow-up of the ALS patient group over time compared to the initial assessment. However, our study observed a positive correlation between BMI and ALSFRS-R Respiratory, Bulbar, and Fine Motor subscores, as well as with the total score. This suggests that patients with a higher BMI tend to have higher ALSFRS-R scores and subscores, indicating fewer clinical disabilities. 

Dardiotis et al. also highlighted in a meta-analysis that BMI is inversely associated with ALS survival [83]. Although we lack survival data in our study, we must address this by noting that BMI was negatively correlated with DPR in our case, indicating that patients with faster progression had a lower BMI. In comparison, those with slower progression had a higher BMI.

Howe et al. studied the post-prandial ghrelin, leptin, and liver-expressed antimicrobial peptide 2 (LEAP 2) levels in ALS patients compared to controls. Their findings showed that ghrelin levels were decreased in ALS patients by more than half compared to controls, evidence of ghrelin resistance. The LEAP 2:ghrelin ratio was greatly increased in ALS patients. They also found out that a lower LEAP2:ghrelin ratio was associated with better functional status assessed using ALSFRS-R. Another result from their study was regarding mortality, proving that higher ghrelin levels and a lower LEAP2:ghrelin ratio had increased earlier death risk. This proves that ghrelin resistance is associated with ALS and might influence the outcome of the patients [84].

Ngo et al. analyzed data from 68 ALS patients using an identical analysis method to ours and found several differences between ALS and controls. Notably, ghrelin, GIP, glucagon, and PP were statistically significantly decreased in ALS as compared to control patients. These data are in contrast with our own as, except for ghrelin, all the others were significantly increased in ALS. This difference could be explained by the fact that the groups of ALS and control in Ngo’s study were not age- and sex-matched and had less severe functional status as their ALSFRS-R score was higher for their patients. Additional research is needed in this specific field in order to assess which variables could influence and cause alterations in metabolic hormones in ALS patients [85].

Our study findings also show decreased ghrelin levels in ALS patients compared to controls, confirming the data from Howe’s study.

### 3.5. Relationship among Metabolic Hormones, Genetic Mutations, and ALS

McDonald et al. observed disrupted glucose homeostasis in SOD1G93A mice, characterized by increased insulin-independent glucose uptake, impaired β-cell function, reduced insulin production, and decreased glucagon sensitivity, suggesting a hormonal influence on ALS progression in this model [15]. Atilano et al. investigated the impact of insulin signaling adjustments on toxicity related to C9orf72 repeat expansions in Drosophila melanogaster and mammalian cells. Flies expressing poly-GR, a dipeptide repeat associated with C9orf72-related neurodegeneration, exhibited reduced insulin signaling pathway components, particularly neuropeptide hormones like Drosophila insulin-like peptides (DILPS). Insulin treatment mitigated various toxic effects induced by C9orf72 repeats, including extending the lifespan, improving locomotor activity, and reducing poly-GR levels. Similar effects were observed in mammalian cells, suggesting a therapeutic potential in targeting insulin signaling pathways for C9orf72-related neurodegenerative diseases [86].

Studies on murine SOD1G93A ALS models revealed elevated glucagon levels following fasting or insulin-induced hypoglycemia despite minimal changes in overall blood glucose levels. This suggests potential dysregulation in glucagon signaling pathways in ALS [15].

Studies in ALS patients and murine models using glucose analogues reveal impaired cellular glucose uptake, evident in the spinal cords and motor cortex of both humans and SOD1G93A mice. Reduced GLUT1 expression in spinal cord astrocytes indicates decreased glucose uptake, while diminished glucose transport is observed in motor cortex synaptosomes during the early symptomatic and end stages of ALS. These findings suggest limitations in glucose transport and altered metabolism in ALS progression [87,88,89,90,91,92].

Limited research on CNS glycolysis in ALS suggests activating the glycolytic pathway could modify disease progression and offer protective benefits. Nicotinamide supplementation, restoring NAD+ levels crucial for glycolysis, enhances survival and glycolytic ATP production in SOD1G93A mice. Elevating NAD+ levels in ALS patients improves functional scores, pulmonary function, and muscular strength. Overexpression of GLUT3 in motor neurons is neuroprotective and improves locomotion in a TDP-43 Drosophila model, suggesting that glycolysis activators could prolong survival in ALS [93,94,95].

Studies in ALS patients and murine models using glucose analogues reveal impaired cellular glucose uptake, evident in the spinal cords and motor cortex of both humans and SOD1G93A mice. Reduced GLUT1 expression in spinal cord astrocytes indicates decreased glucose uptake, while diminished glucose transport is observed in motor cortex synaptosomes during ALS’s early symptomatic and end stages. These findings suggest limitations in glucose transport and altered metabolism in ALS progression [87,88,89,90,91,92].

TDP-43 is a DNA/RNA binding protein that was identified as a component of hyperphosphorylated aggregates in the brain tissue of deceased patients with ALS. Multiple mutations were identified and associated with ALS, proving the role of TDP-43 in familial forms [96,97].

In an experimental study on mice, Ferrer-Donato et al. compared the metabolic profiles of mice with TDP-43 expression versus control. They used a similar analysis method to ours, analyzing total ghrelin, leptin, GIP, GLP-1, insulin, and glucagon plasma levels. They observed lower leptin levels in animals with TDP-43 expression in different ALS stages compared to control group. This shows the connection between TDP-43 expression and ALS, pinpointing leptin as a possible mediator for this relationship, without being able to describe a clear pathway [77].

Araki et al. studied the connection between TDP-43 and glucose intolerance and noticed that patients with ALS presented decreased early-phase insulin secretion as well as the loss of nuclear localization of TDP-43 at the level of the islet pancreatic cells. This proved TDP-43′s role not only in the etiopathogenesis of ALS but in the glucose intolerance development in early phases of ALS [98].

### 3.6. Limitations of the Study

The study’s cross-sectional nature limits causal inference and precludes the establishment of temporal relationships between metabolic hormone levels and ALS progression. Longitudinal studies must elucidate the dynamic interplay between metabolic dysregulation and disease evolution over time.The study’s sample size, while sufficient for identifying correlations, may limit the extension of findings to broader ALS populations. Future studies with larger, more diverse cohorts must validate the observed associations across different demographic and clinical profiles.Despite efforts to control for confounding factors, the influence of unmeasured variables, such as dietary habits, physical activity, and medication use, must be considered. Future research incorporating comprehensive assessments of lifestyle factors and medication histories could better elucidate the independent effects of metabolic hormones on ALS progression.The clinical relevance of correlations requires further investigation.

Further, longitudinal studies are needed to validate these findings and elucidate the potential utility of metabolic biomarkers in clinical practice. Overall, our study contributes to the growing body of evidence implicating metabolic dysfunction in ALS and underscores the importance of considering metabolic factors in the development of novel therapeutic approaches for this devastating disease. By elucidating the role of metabolic pathways in ALS pathogenesis, we may pave the way for more effective treatments that target the underlying molecular mechanisms driving disease progression.

## 4. Materials and Methods

### 4.1. Study Design and Study Population

This prospective, cross-sectional, single-center study aimed to assess the role of metabolic hormones in the progression of ALS. A total of 44 ALS patients and 40 age- and sex-matched healthy controls were recruited between August 2022 and November 2022 among patients diagnosed with ALS in the Neurology Department of Mures University Clinical Emergency Hospital. All ALS patients were diagnosed by experienced clinical ALS neurologists, all enrolled ALS patients were from the Romanian population in Mures County nearby, and there were no racial differences in the study population.

**Inclusion criteria**: (1) ALS patients fulfilled the criteria for diagnosis of definite or probable ALS, according to the El Escorial-Arlie House criteria (with the exclusion of patients with possible ALS) and the Awaji criteria [99,100], (2) aged 25 years or older, and (3) ability to comply with study procedures.

**Exclusion criteria**: (1) diagnosis of other neurodegenerative diseases, (2) presence of other neurological conditions or comorbidities that could confound the assessment of ALS progression, (3) presence of diabetes mellitus or gastrointestinal tract diseases, history of gastrointestinal surgery, and (4) immune-suppressive/immunomodulatory therapy in the last 6 months. All ALS patients were treated with standard therapy for ALS (riluzole 100 mg/day). All participants provided informed consent before enrollment.

The healthy controls were unrelated to the patients, unaffected by neurological disorders or other known diseases, and not underweight (BMI > 18.5 kg/m^2^). 

### 4.2. Data Collection

Baseline demographic data and anthropometric characteristics, including age at study visit (years), gender, age at ALS diagnosis (years), height, body weight, and body mass index (BMI) at diagnosis, were recorded. Clinical data regarding ALS were also obtained: ALS duration at the study date in months (time from the first reported symptom to the date of the first clinical evaluation), ALS diagnostic latency in months (time from symptom onset to ALS diagnosis), the subtype of disease onset (spinal or bulbar), site of ALS onset (cervical vs. lumbar vs. bulbar), site of the body region involved in ALS progression, ALS phenotype (degree of involvement of lower or upper motor neurons), ALS medication therapy, comorbidities, ALS family history, and ALS-related medical interventions, such as percutaneous endoscopic gastrostomy (PEG), non-invasive ventilation (NIV), or tracheostomy. Also, medical histories of metabolic disorders, such as dyslipidemia or diabetes, and gastrointestinal tract diseases were obtained from all study participants during the visit. 

All patients underwent clinical neurological evaluation and electrophysiological (electromyography) evaluation. A comprehensive neurological assessment was conducted, during which the ALS Functional Rating Scale-Revised (ALSFRS-R) was administered. The ALSFRS-R comprises 12 items encompassing bulbar, upper limb, lower limb, and respiratory functions, yielding subscores for each domain: bulbar (ALSFRS-R-B), lower limb (ALSFRS-R-LL), upper limb (ALSFRS-R-UL), and respiratory (ALSFRS-R-R), with 48 being the maximum possible ALSFRS-R score [101]. Baseline ALSFRS-R scores were recorded at the time of diagnosis. 

The rate of ALSFRS-R progression or disease progression rate (DPR) was calculated as the decline in ALSFRS-R score from diagnosis to the date of sample collection, using the formula: 48–[(ALSFRS-R at diagnosis−ALSFRS-R at study visit)/duration of symptoms between onset and study visit (in months)] [102]. To evaluate the differences between fast and slow ALS progression, patients were divided into two groups according to DPR about the median: we defined (1) slow progression with DPR < 0.67 points/month and (2) fast progression with DPR > 0.67 points/month [103].

The distribution of ALS progression was evaluated by examining the spread of muscle atrophy or weakness beyond the initial site of onset, considering clinical, electrophysiological, and anamnestic data. ALS patients were classified into three groups based on the direction of motor neuron degeneration spreading from the onset: (1) horizontal spreading pattern (HSP), extending from the cervical to the contralateral cervical region or from the lumbar to the contralateral lumbar region; (2) vertical spreading pattern (VSP), extending from the cervical or lumbar region to the ipsilateral upper or lower limb, from the bulbar to the lumbar/cervical region, or from the cervical/lumbar to the bulbar region; and (3) crossed spreading pattern (CSP), occurring when clinical or electrophysiological signs spread diagonally to the contralateral side [104].

We applied the King’s College ALS clinical staging system to all patients, stratified into stages: stage 1, symptom onset with involvement of the first region; stage 2A, diagnosis; stage 2B, involvement of the second region; stage 3, involvement of the third region; stage 4A, need for gastrostomy; and stage 4B, need for non-invasive ventilation [105].

The study received approval from the local ethics committees: Research Ethics Board of the Clinical County Emergency Hospital Mures and Research Ethics Board from the University of Medicine, Pharmacy, Sciences and Technology (UMFST) “George Emil Palade” Targu Mures. This research adhered to relevant guidelines and regulations, following the principles stated in the Declaration of Helsinki.

### 4.3. Collection of Biological Samples

Blood samples were collected using EDTA collecting tubes, in the morning after an overnight fast, from each ALS patient or control subject. Plasma samples were treated with a protease inhibitor cocktail. After blood clotting at room temperature for 15 minutes, each sample was centrifuged at 3000 RPM for 15 minutes at 4 degrees Celsius. The supernatant was aliquoted in 0.5 mL sample vials and stored in a −80 degrees Celsius freezer until further use. 

### 4.4. Measurements of Metabolic Hormones’ and Adipokines’ Multiplex Bead-Based Immunoassay

To assess metabolic hormones, serum samples previously stored at −80 °C were transported on ice to the Humoral Immunology Laboratory of the Center for Advanced Medical and Pharmaceutical Research (Targu Mures, Romania). After thawing, samples were vigorously vortexed and processed with a commercially available multiplex bead-based immunoassay according to the manufacturer’s instructions (MILLIPLEX^®^ Human Metabolic Hormone Magnetic Bead Panel kit; Merck-Millipore, Burlington, MA, USA; catalog number HMHEMAG-34K). Samples were then analyzed using the Luminex^®^ xMAP^®^ technology on a properly calibrated and controlled FLEXMAP 3D^®^ analyzer (Luminex Corporation, Austin, TX, USA), and xPONENT^®^ software version 4.3 was used for the analysis of the acquired data. The targets on the metabolic hormones’ assay were insulin, amylin, C-peptide, ghrelin, GIP (gastric inhibitory peptide), GLP-1 active (glucagon-like peptide-1), glucagon, PYY (peptide YY), and PP (pancreatic polypeptide). Targets for the human adipokine assay were leptin, interleukin (IL)-6, MCP-1 (monocyte chemoattractant protein-1), and TNFα (tumor necrosis factor-alpha). All results were expressed in pg/mL.

### 4.5. Statistical Analysis

Statistical analyses were conducted utilizing IBM SPSS Statistics v26 alongside Microsoft Excel 2019. Parametric variables were evaluated through ANOVA testing, with continuous data described using mean and standard deviation (SD). A Spearman correlation coefficient (rho) was employed to assess the correlation between quantitative variables, with significance set at a *p*-value of ≤0.05.

## 5. Conclusions

Decades of research have focused on understanding the pathological mechanisms implicated in ALS, with the hopes of discovering effective treatments for this devastating disease.

The complex relationship between ALS and metabolic hormones shows promising potential treatment targets or pathways that could influence the disease progression and could also improve motor function. The research in this specific field is in an early phase at this point, showing an urge for new studies and data that could support the role of metabolic hormones in ALS prognosis and treatment.

The observed correlations between metabolic hormones and ALS progression provide valuable insights into the complex interplay between metabolic pathways and neurodegeneration in ALS. Our results suggest that metabolic hormones may influence the course of ALS and functional outcomes, warranting further investigation into the underlying mechanisms driving these associations. Targeting metabolic pathways may offer new therapeutic strategies for ALS treatment.

## Figures and Tables

**Figure 1 ijms-25-05059-f001:**
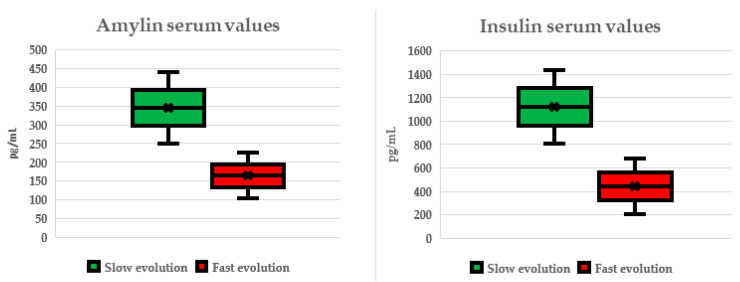
Amylin and insulin serum values’ differences based on the rate of progression.

**Table 1 ijms-25-05059-t001:** ALS and control groups’ characteristics.

	ALS	Control
Number of patients	44	40
Age (mean ± SD) (years)	58.05 ± 12.14	56.97 ± 11.83
Sex ratio	28:16	24:16
BMI	25.07 ± 4.56	NA
FVC(%)	75.73 ± 30.15	NA
ALSFRS-R		
Respiratory subscore	10.95 ± 1.61	NA
Bulbar subscore	9.64 ± 2.63	NA
Gross Motor subscore	6.42 ± 3.23	NA
Fine Motor subscore	7.24 ± 3.42	NA
Total score	33.95 ± 7.92	NA
**ALS SUBTYPE**		
Bulbar	9 (20.45%)	NA
Spinal	35 (79.55%)	NA
**PROGRESSION PATTERN**		
Horizontal pattern	21 (47.73%)	NA
Vertical pattern	23 (52.32%)	NA
**EVOLUTION FORM**		
Slow evolution (DPR < 0.67)	30 (68.18%)	NA
Fast evolution (DPR ≥ 0.67)	14 (31.82%)	NA
**KING’S COLLEGE ALS CLINICAL STAGING**		
Stage 1	0 (0.00%)	NA
Stage 2A	4 (9.09%)	NA
Stage 2B	39 (68.18%)	NA
Stage 3	10 (22.73%)	NA
Stage 4	0 (0.00%)	NA

**Table 2 ijms-25-05059-t002:** Differences in serum values between Control and ALS patients.

	Control	ALS	*p*-Value
Amylin (pg/mL)	156.78 ± 99.86	285.41 ± 359.87	**0.028**
C-peptide (pg/mL)	6932.8 ± 2559.43	5162.95 ± 3792.84	**0.007**
Ghrelin (pg/mL)	63.98 ± 60.78	42.4 ± 55.25	**0.041**
GIP (pg/mL)	4239.65 ± 5171.88	6218.95 ± 4669.2	**0.027**
GLP-1 (pg/mL)	79.03 ± 128.5	132.86 ± 229.91	0.163
Glucagon (pg/mL)	51.1 ± 16.69	72.8 ± 49.63	**0.008**
Insulin (pg/mL)	476.43 ± 662.14	906.76 ± 1226.06	**0.037**
Leptin (pg/mL)	7994.33 ± 4748.2	4662.66 ± 4238.2	**0.000**
MCP-1 (pg/mL)	5524.93 ± 2275	6066.4 ± 4149.51	0.435
PP (pg/mL)	2060.35 ± 2200.31	4992.27 ± 4890.14	**0.000**
PYY (pg/mL)	116.55 ± 195.6	103.98 ± 84.48	0.583

Values of metabolic biomarkers are shown as mean ± standard deviation. GIP, gastric inhibitory peptide; GLP-1, glucagon-like peptide 1; MCP-1, monocyte chemoattractant protein-1; PP, pancreatic polypeptide; PYY, peptide YY.

**Table 3 ijms-25-05059-t003:** Spearman correlations between clinical (ALSFRS, evolution criteria, BMI, FVC) and metabolic hormones’ serum values.

		BMI	Amylin	C-peptide	Ghrelin	GIP	GLP-1	Glucagon	Insulin	Leptin	MCP-1	PP	PYY
DPR	r	−0.223	−0.230	−0.255	0.074	−0.046	0.131	0.142	−0.303	−0.217	0.153	0.018	0.103
	*p*	**0.020**	**0.016**	**0.008**	0.447	0.631	0.173	0.141	**0.001**	**0.023**	0.112	0.850	0.286
ALSFRS-Respiratory	r	0.320	0.260	0.075	0.001	0.165	0.151	−0.069	0.411	0.162	−0.224	0.053	0.044
	*p*	**0.001**	**0.006**	0.437	0.991	0.086	0.115	0.471	**0.000**	0.090	**0.019**	0.584	0.649
ALSFRS-Bulbar	r	0.288	0.237	0.083	−0.122	0.039	0.033	−0.085	0.307	0.105	−0.274	0.116	0.124
	*p*	**0.002**	**0.013**	0.386	0.206	0.685	0.731	0.379	**0.001**	0.275	**0.004**	0.227	0.198
ALSFRS-Gross Motor	r	−0.006	−0.077	−0.125	0.075	0.052	0.077	0.117	−0.021	0.071	−0.089	−0.160	0.085
	*p*	0.951	0.427	0.192	0.435	0.588	0.425	0.222	0.825	0.460	0.356	0.094	0.379
ALSFRS-Fine Motor	r	0.396	0.288	0.093	−0.107	0.124	0.169	0.092	0.341	0.210	−0.148	−0.062	0.079
	*p*	**0.000**	**0.002**	0.332	0.267	0.195	0.078	0.340	**0.000**	**0.028**	0.123	0.522	0.410
ALSFRS-R Total score	r	0.427	0.341	0.048	0.003	0.132	0.205	0.056	0.394	0.169	−0.178	0.018	0.115
	*p*	**0.000**	**0.000**	0.617	0.977	0.170	**0.032**	0.562	**0.000**	0.077	0.063	0.852	0.233
Time of diffusion	r	0.124	0.113	−0.155	0.117	0.006	0.268	0.226	0.137	0.152	−0.298	−0.082	0.112
	*p*	0.198	0.241	0.105	0.225	0.950	**0.005**	**0.018**	0.152	0.113	**0.002**	0.397	0.245
Time of generalization	r	0.115	0.117	−0.163	0.128	0.002	0.263	0.221	0.138	0.130	−0.297	−0.076	0.110
	*p*	0.233	0.224	0.089	0.181	0.985	**0.006**	**0.020**	0.152	0.177	**0.002**	0.429	0.252
BMI	r	1.000	0.306	0.222	−0.116	−0.029	0.225	−0.035	0.340	0.590	−0.139	−0.077	−0.054
	*p*		**0.001**	**0.020**	0.229	0.761	**0.018**	0.715	**0.000**	**0.000**	0.148	0.424	0.578
FVC(%)	r	0.281	0.245	0.134	0.145	−0.015	0.051	−0.083	0.340	0.182	−0.245	−0.003	0.151
	*p*	**0.003**	**0.010**	0.163	0.129	0.876	0.598	0.391	**0.000**	0.056	**0.010**	0.973	0.116
Amylin	r	0.306	1.000	0.529	0.146	0.493	0.104	0.146	0.748	0.237	0.020	0.300	0.091
	*p*	**0.001**		**0.000**	0.127	**0.000**	0.277	0.128	**0.000**	**0.013**	0.838	**0.001**	0.345
C-peptide	r	0.222	0.529	1.000	0.010	0.298	−0.200	−0.125	0.606	0.254	−0.033	0.178	0.024
	*p*	**0.020**	**0.000**		0.920	**0.002**	**0.036**	0.193	**0.000**	**0.008**	0.734	0.063	0.806
Ghrelin	r	−0.116	0.146	0.010	1.000	0.205	0.148	0.317	0.098	−0.095	0.026	0.321	0.259
	*p*	0.229	0.127	0.920		**0.032**	0.122	**0.001**	0.307	0.324	0.784	**0.001**	**0.006**
GIP	r	−0.029	0.493	0.298	0.205	1.000	0.212	0.316	0.482	−0.048	0.054	0.569	0.234
	*p*	0.761	**0.000**	**0.002**	**0.032**		**0.026**	**0.001**	**0.000**	0.617	0.576	**0.000**	**0.014**
GLP-1	r	0.225	0.104	−0.200	0.148	0.212	1.000	0.581	0.160	0.068	−0.280	0.358	0.483
	*p*	**0.018**	0.277	**0.036**	0.122	**0.026**		**0.000**	0.094	0.479	**0.003**	**0.000**	**0.000**
Glucagon	r	−0.035	0.146	−0.125	0.317	0.316	0.581	1.000	0.074	−0.129	0.043	0.484	0.567
	*p*	0.715	0.128	0.193	**0.001**	**0.001**	**0.000**		0.445	0.181	0.654	**0.000**	**0.000**
Insulin	r	0.340	0.748	0.606	0.098	0.482	0.160	0.074	1.000	0.299	−0.028	0.305	0.179
	*p*	**0.000**	**0.000**	**0.000**	0.307	**0.000**	0.094	0.445		**0.002**	0.772	**0.001**	0.062
Leptin	r	0.590	0.237	0.254	−0.095	−0.048	0.068	−0.129	0.299	1.000	−0.132	−0.112	−0.026
	*p*	**0.000**	**0.013**	**0.008**	0.324	0.617	0.479	0.181	**0.002**		0.168	0.242	0.786
MCP-1	r	−0.139	0.020	−0.033	0.026	0.054	−0.280	0.043	−0.028	−0.132	1.000	−0.001	−0.143
	*p*	0.148	0.838	0.734	0.784	0.576	**0.003**	0.654	0.772	0.168		0.995	0.137
PP	r	−0.077	0.300	0.178	0.321	0.569	0.358	0.484	0.305	−0.112	−0.001	1.000	0.421
	*p*	0.424	**0.001**	0.063	**0.001**	**0.000**	**0.000**	**0.000**	**0.001**	0.242	0.995		**0.000**
PYY	r	−0.054	0.091	0.024	0.259	0.234	0.483	0.567	0.179	−0.026	−0.143	0.421	1.000
	*p*	0.578	0.345	0.806	**0.006**	**0.014**	**0.000**	**0.000**	0.062	0.786	0.137	**0.000**	

**Table 4 ijms-25-05059-t004:** Significant differences in ALS patients with slow evolution compared to fast evolution.

	Slow Evolution	Fast Evolution	*p*
Amylin (pg/mL)	343.86 ± 409.63	162.96 ± 178.86	0.014
Insulin (pg/mL)	1116.35 ± 1365.25	437.51 ± 685.87	0.007

## Data Availability

The data presented in this study are available on request from the corresponding author due to privacy.

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
