# Peer review of "Exploring the Role of Metabolic Hormones in Amyotrophic Lateral Sclerosis"

_ijms, 2024, doi:10.3390/ijms25105059_

Round 1

Reviewer 1 Report

Comments and Suggestions for Authors

This manuscript reports the findings of a cross-sectional study where blood serum concentrations of metabolic hormones in ALS patients were compared to controls and association with severity of disease, progression patterns and clinical outcomes were examined. Key findings include increased insulin and amylin as well as several other hormones which showed strong correlations with disease progression and functional status.  The authors also found differences between some metabolic hormones in their subgroup analysis, with males showing higher amylin and glucagon than females whilst leptin showed the reverse.

I am concerned that some of data are different to what others have found. As examples, the authors found increased levels of amylin, insulin, GIP, glucagon, and PP whereas C-peptide was reduced in ALS. Insulin and amylin showed the most correlation with disease progression and functional outcomes but these major hormones were not shown to be altered in ALS patients in the Ngo et al (2015) study who used a larger ALS cohort (68 patients). Furthermore, Ngo et al reported no change in glucagon and PYY and the levels of PP were reduced in their study which are in contrast to the present discoveries. Ngo et al also found increased IL-6 and IL-8, which are different to the present findings. Both studies used similar approaches for detection and measurements of metabolic hormones i.e., multiplex immunoassays so one would expect similar outcomes. Have there been recent studies that reported similar trends in metabolic hormone expression in ALS sera/plasma using similar or more sensitive assays?     

The authors should include more discussion on TDP-43 in light of their discoveries of altered metabolic hormones in ALS patients as the detrimental effects of TDP-43 are well known to impact not just lipid metabolism but also other energy producing pathways that encompass metabolic cascades such as glycolysis. TDP-43 is a known regulator of insulin secretion and the loss of such may be linked to its role in insulin intolerance in early-stage ALS patients (Araki et al., 2019). Consistent findings in SOD1 and TDP-43 mouse models also suggest that TDP-43 contributes majorly to metabolic dysfunction and energy deficits in ALS pathogenesis. Have there been recent studies that have investigated changes in metabolic hormones in TDP-43 ALS models?

The authors discuss that patients with ALS have certain features of T2 diabetes, including hyperinsulinemia and glucose intolerance and that this is protective against ALS. They should also include the Mariosa et al (2015) study that reported insulin dependent T1 diabetes appears to increase the risk of ALS (odds ratio 5.38). An explanation for the contrasting effects of T1 and T2 diabetes on ALS should be included. Also important to consider that while insulin resistance (a hallmark of T2 diabetes) may have protective effects against ALS, the protective effects could be secondary to other factors (e.g., environmental/genetic) that contributes to the onset of T2 diabetes.

The authors should include the recent paper by Howe et al (2023) in their discussion of Ghrelin and ALS. They showed ghrelin correlated with diagnostic delay and that patients with higher postprandial ghrelin levels and lower LEAP2:ghrelin ratios had higher risk of earlier death, suggesting that altered levels of certain metabolic hormones might alter disease outcome.

Higher serum pancreatic polypeptide (PP) in the ALS patients is a notable finding in this study, with levels being more than doubled the control values (p=0.000). It would be interesting to have the authors discuss this this further.

Minor comments:

There are a number of typographical errors in the text:

“ALSFSR-R” appears in Table 3 and throughout the manuscript – please change “ALSFSR-R” to “ALSFRS-R.”

Please remove the duplications in Page 9 lines 321 to 367 – these are duplication of lines 308 to 313 on page 8.

No units have been included in Table 2 – please add pg/ml to the Table.

Author Response

Response to reviewer 1

This manuscript reports the findings of a cross-sectional study where blood serum concentrations of metabolic hormones in ALS patients were compared to controls and association with severity of disease, progression patterns and clinical outcomes were examined. Key findings include increased insulin and amylin as well as several other hormones which showed strong correlations with disease progression and functional status.  The authors also found differences between some metabolic hormones in their subgroup analysis, with males showing higher amylin and glucagon than females whilst leptin showed the reverse.

I am concerned that some of data are different to what others have found. As examples, the authors found increased levels of amylin, insulin, GIP, glucagon, and PP whereas C-peptide was reduced in ALS. Insulin and amylin showed the most correlation with disease progression and functional outcomes but these major hormones were not shown to be altered in ALS patients in the Ngo et al (2015) study who used a larger ALS cohort (68 patients). Furthermore, Ngo et al reported no change in glucagon and PYY and the levels of PP were reduced in their study which are in contrast to the present discoveries. Ngo et al also found increased IL-6 and IL-8, which are different to the present findings. Both studies used similar approaches for detection and measurements of metabolic hormones i.e., multiplex immunoassays so one would expect similar outcomes. Have there been recent studies that reported similar trends in metabolic hormone expression in ALS sera/plasma using similar or more sensitive assays?     

Response: Unfortunately, there are very few studies that tried to assess the metabolic hormone expression in ALS patients, thus making it difficult to compare to a broad range of literature results or to make a consensus regarding the metabolic hormone expression trends. Although Ngo et al. results are somehow in contrast to our own results, there is a dire need for additional research in this specific field, in order to properly compare our data with other published data.

Referring to Ngo et al. findings regarding IL-6 and IL-8 we also did multiplex immunoassay for cytokines but we chose to publish that data in a different article, and we chose not to add it to this article as well as it can be considered as duplicate research data.  As for the difference in results between our study and Ngo’s we pointed out that their group of ALS patients were not age and sex matched and also, had less severe ALS forms as suggested by the higher ALSFRS-R. This means that we cannot fully compare our results with theirs without expecting differences.

The authors should include more discussion on TDP-43 in light of their discoveries of altered metabolic hormones in ALS patients as the detrimental effects of TDP-43 are well known to impact not just lipid metabolism but also other energy producing pathways that encompass metabolic cascades such as glycolysis. TDP-43 is a known regulator of insulin secretion and the loss of such may be linked to its role in insulin intolerance in early-stage ALS patients (Araki et al., 2019). Consistent findings in SOD1 and TDP-43 mouse models also suggest that TDP-43 contributes majorly to metabolic dysfunction and energy deficits in ALS pathogenesis. Have there been recent studies that have investigated changes in metabolic hormones in TDP-43 ALS models?

Response: We have added additional discussions regarding TDP-43, including possible interactions with metabolic hormones in experimental animal studies. We also followed your suggestion regarding TDP-43 and insulin intolerance role in early stage ALS patients and added additional information.

The authors discuss that patients with ALS have certain features of T2 diabetes, including hyperinsulinemia and glucose intolerance and that this is protective against ALS. They should also include the Mariosa et al (2015) study that reported insulin dependent T1 diabetes appears to increase the risk of ALS (odds ratio 5.38). An explanation for the contrasting effects of T1 and T2 diabetes on ALS should be included. Also important to consider that while insulin resistance (a hallmark of T2 diabetes) may have protective effects against ALS, the protective effects could be secondary to other factors (e.g., environmental/genetic) that contributes to the onset of T2 diabetes.

Response: Per your suggestion, we added a new paragraph discussing type 1 and type 2 possible associations with ALS.

The authors should include the recent paper by Howe et al (2023) in their discussion of Ghrelin and ALS. They showed ghrelin correlated with diagnostic delay and that patients with higher postprandial ghrelin levels and lower LEAP2:ghrelin ratios had higher risk of earlier death, suggesting that altered levels of certain metabolic hormones might alter disease outcome.

Response: We added a new paragraph discussing the article you suggested and compared it to our findings.

Higher serum pancreatic polypeptide (PP) in the ALS patients is a notable finding in this study, with levels being more than doubled the control values (p=0.000). It would be interesting to have the authors discuss this this further.

Response: We added a new paragraph including PP findings, as suggested.

Minor comments:

There are a number of typographical errors in the text:

“ALSFSR-R” appears in Table 3 and throughout the manuscript – please change “ALSFSR-R” to “ALSFRS-R.”

Response: We corrected the typographic errors.

Please remove the duplications in Page 9 lines 321 to 367 – these are duplication of lines 308 to 313 on page 8.

Response: We removed duplications.

No units have been included in Table 2 – please add pg/ml to the Table.

Response: We added units in Table 2 and also in Table 4.

Reviewer 2 Report

Comments and Suggestions for Authors

This paper clinically examines the relationship between various metabolic abnormalities in Amyotrophic lateral sclerosis (ALS). Various theories have been proposed to explain the onset of ALS, but it is not completely understood, and this study takes a different approach. Regarding this report, there are no objections to the methodology or number of patients.

On the other hand, in the discussion section, we discuss each possibility for the differences obtained, but regarding BMI, for example, it does not mean that patients with a high BMI develop the disease, nor vice versa. Differences in blood sugar control hormones have also been observed between ALS patients and normal subjects, but it is not clear from this study whether this finding is the root cause of ALS. It is suggested that there is a high possibility that abnormalities will be recognized in these values after developing ALS.

It is necessary to summarize how measuring these metabolic abnormalities will help in the early detection and treatment of ALS patients in the future.

Author Response

Response to reviewer 2

This paper clinically examines the relationship between various metabolic abnormalities in Amyotrophic lateral sclerosis (ALS). Various theories have been proposed to explain the onset of ALS, but it is not completely understood, and this study takes a different approach. Regarding this report, there are no objections to the methodology or number of patients.

On the other hand, in the discussion section, we discuss each possibility for the differences obtained, but regarding BMI, for example, it does not mean that patients with a high BMI develop the disease, nor vice versa. Differences in blood sugar control hormones have also been observed between ALS patients and normal subjects, but it is not clear from this study whether this finding is the root cause of ALS. It is suggested that there is a high possibility that abnormalities will be recognized in these values after developing ALS.

Response: We added new additional paragraphs including the relationship between blood sugar and insulin resistance. We explained that these findings are not the root cause of ALS but are risk factors and are associated with higher risk of ALS incidence.

It is necessary to summarize how measuring these metabolic abnormalities will help in the early detection and treatment of ALS patients in the future.

Response: We added additional data both in discussion and conclusion regarding metabolic abnormalities and their role in the future of ALS treatment.

Reviewer 3 Report

Comments and Suggestions for Authors

The manuscript present a cross-sectional study of 44 ALS patients in order to investigate the relationship between metabolic hormones and disease progression. 

My comments are the following:

·         The abstract and introduction are concise.

·         The results present the findings detailed.

·         Discussion is extremely long and very hard to follow. It should be shortened and only the relevant information to the findings should be given. Subchapters should be formed. Repetitions should be avoided. Maybe the discussion would profit from a figure of the possible pathological pathways. The reference list is too long also, try to find the most relevant publications to the topic. 

Author Response

Response to reviewer 3

The manuscript present a cross-sectional study of 44 ALS patients in order to investigate the relationship between metabolic hormones and disease progression. 

My comments are the following:

  • The abstractand introduction are concise.
  • Theresults present the findings detailed.
  • Discussion is extremely long and very hard to follow. It should be shortened and only the relevant information to the findings should be given. Subchapters should be formed. Repetitions should be avoided. Maybe the discussion would profit from a figure of the possible pathological pathways. The reference list is too long also, try to find the most relevant publications to the topic. 

Response: Thank you for your suggestions. We added subchapters for a better read flow. We removed any duplicates of repetitions found.

Round 2

Reviewer 1 Report

Comments and Suggestions for Authors

The authors have satisfactorily responded to all my comments

Reviewer 2 Report

Comments and Suggestions for Authors

The authors of this paper have made appropriate corrections to the previous points. There are no further comments.